# Analysis of Intestinal Microbial Diversity of Four Species of Grasshoppers and Determination of Cellulose Digestibility

**DOI:** 10.3390/insects13050432

**Published:** 2022-05-05

**Authors:** Jing Bai, Yao Ling, Wen-Jing Li, Li Wang, Xiao-Bao Xue, Yuan-Yi Gao, Fei-Fei Li, Xin-Jiang Li

**Affiliations:** The Key Laboratory of Zoological Systematics and Application, School of Life Sciences, Institute of Life Sciences and Green Development, Hebei University, Baoding 071002, China; baijing2566@163.com (J.B.); lingyao0112@163.com (Y.L.); wenjing12022021@163.com (W.-J.L.); hbuwangl@163.com (L.W.); xuexiaobao2021@163.com (X.-B.X.); gaoyy0302@163.com (Y.-Y.G.); lifeiyazhou@163.com (F.-F.L.)

**Keywords:** grasshopper, gut microbiome, microbial diversity, cellulose digestibility, 16S rRNA

## Abstract

**Simple Summary:**

Grasshoppers are typical phytophagous pests, which prefer eating monocotyledons with more cellulose and hemicellulose. Due to its large appetite and high utilization rate, the intestinal contents of grasshoppers have the potential to be developed into a bioreactor, which can be applied to improve straw utilization efficiency in the future. The digestive tract of grasshoppers is a complex ecosystem, inhabited by a large number of microorganisms. The existence of these microorganisms enables grasshoppers to have high decomposition and utilization of plant fibers. However, there are few reports on the microflora structure and diversity of the digestive tract of grasshoppers. In this study, the diversity of symbiotic bacteria in the intestinal tract of four species of grasshoppers, namely *Acrida cinerea*, *Trilophidia annulata*, *Atractomorpha sinensis* and *Sphingonotus mongolicus*, was studied by using the method of constructing a 16S rRNA gene library and Illumina Miseq sequencing technology. At the same time, the digestibility of cellulose and hemicellulose of the four species of grasshoppers were determined and the relationship between digestibility and intestinal microbial diversity was analyzed. This study provided basic data for the development of the digestible bioreactor of cellulose and hemicellulose, which may provide a new idea for degrading straw.

**Abstract:**

Grasshoppers (Insecta, Orthoptera, Acridoidea) are a large group of agricultural and animal husbandry pests. They have a large food intake with high utilization of plants fibers. However, the composition of the grasshopper gut microbial community, especially the relationship between gut microbial community and cellulose digestibility, remains unclear. In this research, 16S rRNA gene sequences were used to determine the intestinal microbial diversity of *Acrida cinerea*, *Trilophidia annulata*, *Atractomorpha sinensis* and *Sphingonotus mongolicus*, and Spearman correlation analysis was performed between the intestinal microbes of grasshoppers and the digestibility of cellulose and hemicellulose. The results showed that Proteobacteria was the dominant phylum and *Klebsiella* was the dominant genus in the guts of the four species of grasshoppers; there was no significant difference in the species composition of the gut microbes of the four species of grasshoppers. Spearman correlation analysis showed that *Brevibacterium* and *Stenotrophomonas* were significantly correlated with cellulose digestibility. *Brevibacterium*, *Clavibacter*, *Microbacterium* and *Stenotrophomonas* were significantly associated with hemicellulose digestibility. Our results confirmed that the gut microbes of grasshoppers were correlated with the digestibility of cellulose and hemicellulose, and indicated that grasshoppers may have the potential to develop into bioreactors, which can be applied to improve straw utilization efficiency in the future.

## 1. Introduction

Insects are the largest group of animals, are widely distributed in the world and have a long evolutionary history [1]. The insect gut is the place where various nutrients and metabolic wastes are exchanged with the external environment, and it hosts a large number of microorganisms. There is a co-evolutionary relationship between intestinal microorganisms and the host, and the core microorganisms of the intestinal tract are different among species. Hongoh et al. proved that there is a certain relationship between the phylogeny of different species of termite gut symbiotic microbes and termite species, indicating that termite gut microbes have a co-evolutionary relationship with the host [2]. Different feeding habits and living environments affect the intestinal structure and function of insects, and vice versa [3]. Aziz et al. compared the similarities and differences of intestinal microorganisms of three grasshoppers by biochemical and molecular research methods, purified and isolated the bacteria, and attributed the different reasons of microorganisms to the different geographical environment [4]. Moreover, Lavy et al. summarized the research on the diversity of intestinal microorganisms of desert grasshoppers and migratory grasshoppers, including *Locusta pardalina*, *Dociostarus marocanus* and *Callipamus Italicus*, and proved that the composition of bacterial colonies in the digestive tract of grasshoppers is largely affected by their specific anatomical structure [5]. In the gut of insects, many types of microorganisms are available, which can be divided into resident microflora and passing microflora. The resident microflora occupies a certain position and performs a specific function in the gut, which exists in the hosts for a long time along with their growth and development. They are closely related to the hosts, and their population is maintained in a dynamic equilibrium mechanism. The passing microflora is transient in the hosts and is excreted as metabolic wastes [6]. Intestinal microbes provide nutrients to the host [7], help digest stubborn food components [8], protect the host from predators [9], parasites and pathogens [10], affect the efficiency of disease vectors [11] and even affect the mating and reproductive system of the host [12]. The microbial community in the insect digestive tract plays an important role, which not only ensures the orderly operation of an insect body, but also has an important impact on human production and life in medicine, agriculture and ecology. Therefore, insect intestinal microorganisms are ideal materials for studying relevant evolutionary mechanisms, as well as a huge microbial resource bank to find key microorganisms with biological functions, studying their functional mechanisms and ultimately applying it to production practice.

A large number of crop stalks cannot be effectively used in the world. Incineration will cause environmental pollution and waste resources. One of the most difficult problems that human beings face is how to solve the problem of crop straw recycling. Grasshoppers are widely distributed, have large appetites, strong reproductive capacity and migrate fast. Swarms of grasshoppers can do great harm to crops or pastures. Termites, grasshoppers and longicorn beetles feed on cellulose and contain a variety of symbiotic bacteria that degrade cellulose, including Enterobacteriaceae, *Bacteroides*, *Staphylococcus*, *Streptococcus* and *Bacillus*, and its degradation capacity of lignin model compounds is about 20–100%, which is 30–40% higher than that of large herbivores [13]. Su et al. studied 16 species of grasshopper intestinal symbiotic bacteria through DEEG, and the results showed that cellulolytic enzymes and intestinal microbial communities may reflect the relationship between different species of grasshopper and their feeding patterns [14]. The core gut bacteria of *Cytrotrachelus Buqueti*, a bamboo nose beetle, have carbohydrate-active enzymes that are key to lignocellulosic degradation and are used to break down bamboo cell walls, thereby contributing to the growth of host insects [15]. There are few studies on the relationship between cellulose digestibility and gut microbial community structure in grasshoppers. Grasshoppers maybe have the potential to be developed into bioreactors, which can be applied to improve straw utilization efficiency in the future.

*Acrida cinerea* Thunberg, 1815 (Ac), *Trilophidia annulata Thunberg*, 1815 (Ta), *Atractomorpha sinensis* Bolivar, 1905 (As) and *Sphingonotus mongolicus* Saussure, 1888 (Sm) are used in the study. The 16S rRNA gene sequences of four species of grasshopper intestinal bacteria were sequenced by the paired-end sequencing method and construction of a small fragment library. The intestinal microbial diversity of these species was further analyzed. The digestibility of cellulose and hemicellulose of four species of grasshoppers were determined by moss black phenol colorimetry and anthrone colorimetry, respectively. Furthermore, the relationship between digestibility and intestinal microbial diversity was analyzed. It provides a new thought for the green utilization of crop straw, which has important theoretical and practical significance.

## 2. Materials and Methods

### 2.1. Specimen Collection

Adults of *Acrida cinerea* Thunberg, 1815 (Ac), *Trilophidia annulata* Thunberg, 1815 (Ta), *Atractomorpha sinensis* Bolivar, 1905 (As) and *Sphingonotus mongolicus* Saussure, 1888 (Sm), were collected from Baoding City, Hebei Province, China in July–October 2018 (Table 1). They had the same living environment.

### 2.2. Sample Treatment

Grasshoppers collected from the wild were classified and placed in different cages on an empty stomach for 2 days, so that their intestines were emptied of feces. The grasshoppers were immersed in 70% ethanol solution for 5 min to sterilize the bacteria on the grasshoppers’ surface. In the ultra-clean working table, the bodies were placed on the sterilizing glass plate. The legs and wings were cut off by sterilized scissors and the bodies were cut from the anus to the chest along the abdomen. The surface of the bodies was cut open with sterilized dissecting needles, and the guts were removed with sterilized forceps. The guts were put into sterilized 1.5 mL EP tubes. Each tube contained a sample of five female and five male guts of the same species. There were three samples of each species. Intestinal dissection procedures were performed on ice.

### 2.3. Extraction of Total DNA from the Intestinal Contents

The PowerSoil DNA Isolation Kit was used to extract total DNA from the guts of grasshoppers. The common primers—338F (5′-ACTCCTACGGGAGCAGCA-3′) and 806R (5′-GGACTACHVGGGTWTCTAAT-3′)—in the V3 + V4 region of bacterial 16S rDNA were used as amplification primers [16]. Sequencing adapters were added to the primer ends to perform PCR. Target region PCR was performed in a total reaction volume of 10 µL: KOD FX Neo Buffer, 5.0 µL; DNA template, 50 ± 0 ng; primer1 (10 mmol/L), 0.3 µL; primer2 (10 mmol/L), 0.3 µL; dNTP, 2.0 µL; KOD FX Neo (5 U/mL), 0.2 µL; and constant volume to 10 µL with ddH_2_O. After an initial denaturation at 95 °C for 5 min, amplification was performed with 25 cycles of incubations for 30 s at 95 °C, 30 s at 50 °C and 40 s at 72 °C, followed by a final extension at 72 °C for 7 min. Then, Solexa PCR was performed in a total reaction volume of 20 µL: 2 × Q5 HF MM, 10.0 µL; Target region PCR product (100 ng/mL), 5 µL; primer1 (2 mmol/L), 2.5 µL; primer2 (2 mmol/L), 2.5 µL. After an initial denaturation at 98 °C for 30 s, amplification was performed with 10 cycles of incubations for 10 s at 98 °C, 30 s at 65 °C and 30 s at 72 °C, followed by a final extension at 72 °C for 5 min. The amplified products were then purified and recovered using 1.8% agarose gel electrophoresis. The products were purified, quantified and homogenized to form sequencing libraries. The qualified sequencing libraries were sequenced with Illumina HiSeq 2500 (2 × 250 pairedends) at Biomarker Technologies Corporation, Beijing, China. This process was completed by Beijing Biomarker Technologies Co., Ltd.

### 2.4. Microbial Diversity Analysis

The original data were paired by FLASH 1.2.7 (overlap > 10 bp, false match rate ≤ 0.2) [17]. The paired raw reads were filtered by Trimmomatic v0.33 [18]. The Clean Reads without primer sequences were obtained by using Cutadapt 1.9.1 to identify and remove primer sequences. After the Clean Reads of each sample were paired, the Usearch v10 was used for length filtering. After removing chimeras by UCHIME v8.1 [19], the tags sequence with a higher quality was finally obtained. With Silva as the reference database, the feature sequences were annotated by using a Naive Bayes classifier. With 0.005% of all sequences as the filtering threshold [20], the sequences were clustered by Operational taxonomic unit (OTU) at the 97% similarity level [21], and OTU was taxonomically annotated. This process was completed by Beijing Biomarker Technologies Co., Ltd.

### 2.5. Digestibility of Cellulose and Hemicellulose

The grasshoppers collected from the wild were kept in cages and were fed wheat (*Triticum aestivum* Linnaeus, 1753) seedlings. After consecutively feeding for 3 days (no dung was collected during the period), grasshoppers were fasted for 2 days. At the beginning of the formal experiment, the grasshoppers were fed with wheat seedlings regularly and quantitatively, and feces were collected at the same time. The intake and excretion of grasshoppers were recorded in a day. The grasshoppers were fed every day at 9 am and 5 pm, and the feces excreted by the grasshoppers and the remaining wheat seedlings were collected the next morning. The formal experiment lasted for a week. The content of cellulose and hemicellulose in feces and wheat were determined by moss black phenol colorimetry and anthrone colorimetry, respectively. The following formulas were used to calculate the cellulose digestibility and hemicellulose digestibility of grasshoppers. Refer to Wang for specific methods [22].
$$\mathrm{Cellulose}\;\left(\mathrm{hemicellulose}\right)\;\mathrm{content}\;\left(\%\right)=\left(c\times 240\times 10^{-3}\;L\times 0.9\left(0.88\right)\right)/m\times \mathrm{dilution}\;\mathrm{mutiple}\times 100\%$$
Cellulose (hemicellulose) digestibiliy (%)=(a−b)/a×100%

Note: c is the sugar concentration (g/L) calculated according to the standard curve, 240 is the total volume of sample solution (mL), m is the weighed sample mass (g), 0.9 and 0.88 are coefficients. a is amount of cellulose fed on wheat seedlings (g), b is fecal cellulose content (g).

### 2.6. Correlation between Digestibility and Intestinal Microbial Diversity

The LefSe analysis and Spearman analysis were performed using R and the Psych, Pheatmap and reshape2 package on the Biomarker Cloud Platform (Biomarker Biotechnology Co., Beijing, China) [23]. The correlation between cellulose digestibility and intestinal microbial diversity of grasshoppers was established.

## 3. Results

### 3.1. Analysis of the Gut Microbiota Diversity and Bacterial Composition between Four Species of Grasshoppers

In the guts of *Acrida cinerea* (Ac), *Trilophidia annulata* (Ta), *Atractomorpha sinensis* (As) and *Sphingonotus mongolicus* (Sm), a total of 858,719 pairs of reads were obtained from 12 samples; 758,316 Clean Reads were produced after quality control and splicing of paired-end reads.

A total of 7 phyla, 12 classes, 21 orders, 42 families and 55 genera were annotated for the four species of grasshoppers (Table 2). As shown in Figure 1A, as the sample number increased, the cumulative curve and the common quantity curve tended to be flat, which demonstrates that the new and common species detected in the sample were both approaching saturation, and the sample size was sufficient and could be used for diversity and abundance analysis. The Shannon, Chao1 and ACE indices were used to express the α-diversity of the microorganisms in the samples. As shown in Table 3, the coverage of the 12 samples was relatively high, reaching 99.97~99.99%. The above results showed that the sequencing data were reasonable and that the vast majority of bacteria in the samples were detected.

The average value of each index of the three samples from the same species was calculated (Table 3) and then used to compare and analyze the α-diversity among the different species. In a community, species diversity was affected by the richness and evenness of the species. The ACE index and Chao1 index reflected the species richness. The Chao1 index (Figure 1B) and ACE index (Figure 1C) of *Acrida cinerea* (Ac) was the highest, and *Atractomorpha sinensis* (As) was the lowest. The Shannon index reflected the species diversity. The Shannon index (Figure 1D) of *Sphingonotus mongolicus* (Sm) was the highest, and *Atractomorpha sinensis* (As) was the lowest. In the guts of the four species of grasshoppers, the species richness in descending order was *Acrida cinerea* (Ac), *Sphingonotus mongolicus* (Sm), *Trilophidia annulate* (Ta) and *Atractomorpha sinensis* (As). The species diversity in descending order was *Sphingonotus mongolicus* (Sm), *Trilophidia annulate* (Ta), *Acrida cinerea* (Ac) and *Atractomorpha sinensis* (As).

PCoA was principal coordinate analysis, which can further display the differences in species diversity between samples. In Figure 2, the closer the distance of the graphic indicates that the samples are more similar. Except for one sample of *Acrida cinerea* (Ac), the other samples of *Acrida cinerea* (Ac) and *Atractomorpha sinensis* (As) were close to each other, indicating that the samples of *Acrida cinerea* (Ac) and *Atractomorpha sinensis* (As) were similar. The three samples of *Sphingonotus mongolicus* (Sm) were close in distance and similar, and there was no obvious difference between individuals, but the three samples of *Trilophidia annulate* (Ta) were scattered, and the inter-individual differences of *Trilophidia annulate* (Ta), were larger than those of the other three species of grasshoppers. This result was only affected by the presence or absence of species, not by species abundance. The gut microbial composition of the four grasshopper species was different, but the difference was not significant (*p* > 0.05). This may be related to the same living environment of the four species of grasshoppers.

At the phylum level (Figure 3A), a total of 7 phyla were obtained from 12 samples. According to the abscissa in Figure 3A, from left to right: *Acrida cinerea* (Ac), *Atractomorpha sinensis* (As), *Sphingonotus mongolicus* (Sm), *Trilophidia annulate* (Ta). Proteobacteria accounted for the highest proportion at 96.62, 99.23, 87.66 and 64.25%, respectively. The rest were Firmicutes (2.36, 0.11, 7.43, 25.29%), Cyanobacteria (0.02, 0.58, 0.08, 10.19%), Actinomycetes (0.14, 0.03, 4.87, 0.05%), Bacteroides (0.78, 0.06, 0.94, 0.76%), Tenericutes (0.08%, 0, 0, 0.14%) and Fusobacteria (0, 0, 0.10%, 0). Among them, Proteobacteria was the absolute dominant phylum. Further, Firmicutes had a larger proportion in the *Sphingonotus mongolicus* (Sm) and *Trilophidia annulate* (Ta) guts than the other two species of grasshoppers. At the family level (Figure 3B), Enterobacteriaceae was ubiquitous in most samples. *Trilophidia annulate* (Ta) had the lowest relative abundance of Enterobacteriaceae. However, the relative abundance of Streptoceccaceae in the *Trilophidia annulate* (Ta) group was significantly higher than that in the other three groups. The absolute dominant bacteria of the four species of grasshoppers at the phylum level and the family level were similar.

Figure 4 combined the UPGMA cluster tree with the abundance histogram of each sample at the genus level. The similarity of species composition and abundance o among different samples could be intuitively judged. A total of 55 genera were annotated from 12 samples. The abundance histogram on the right showed the top ten genera with relative abundance greater than 1% in the intestines of four species of grasshoppers. However, according to the abundance histograms, the four species of grasshoppers differed at the genus level. There were *Klebsiella* and *Staphylococcus* in *Acridia chinensis* (Ac), *Klebsiella* and *Wolbachia* in *Atractomorpha sinensis* (As), *Klebsiella*, *Acinetobacter*, *Pantoea*, *Enterococcus*, *Staphylococcus*, *Stenotrophomonas*, *Microbacterium*, *Brevibacterium* and *Corynebacterium* in *Sphingonotus mongolicus* (Sm), *Klebsiella*, *Lactococcus* and *Enterococcus* in *Trilophidia annulate* (Ta). *Klebsiella* was the dominant genus shared by four species of grasshoppers. Compared with other genus, *Klebsiella* had the largest proportion, that is, the largest relative abundance. *Morganella* was unique to *Sphingonotus mongolicus* (Sm). Among the three samples of *Sphingonotus mongolicus* (Sm), it was only detected in Sm1, and the abundance was less than 0.01%. The cluster tree on the left shows the species composition in Figure 4 are most similar in samples of As1 and As3 and samples of Ac2 and Ta3. Ta1 and Ta2 had the obvious difference with other samples. Excluding Ta1 and Ta2, the difference between Sm1 and other samples was the most obvious. The difference in biodiversity among the three samples of the same species may be related to the difference in collection time and individual grasshopper.

The distance matrix was calculated by the weighted UniFrac method. A sample heat map was drawn by the R language tool. The heat map is a picture that uses color to represent the differences between samples. The color gradient from blue to red indicated that the distance between the samples was from close to far. Differences between two samples can be visually seen based on changes in the color gradient. The results were shown in Figure 5. The difference between Ta1 and other samples was red. The difference between Ta2 with Sm1 and other samples was between red and blue. The differences of the rest samples were blue, indicating that the microbial diversity and abundance were slight but insignificantly different among most samples. It was consistent with PCoA results (*p* > 0.05).

In order to find biomarkers with statistical differences between different groups, we used linear discriminant analysis (LDA) effect size (LEfSe) to screen out different taxa at various levels (kingdom, phylum, class, order, family, genus, species) between different groups based on a standard LDA value greater than four (Figure 6). Biomarkers are molecules found in the body that indicate a specific biological condition. The biomarkers with LDA Scores greater than the set value of 4.0 were displayed and only screened in the guts of *Atractomorpha sinensis* (As) and *Sphingonotus mongolicus* (Sm). The LDA Scores of family Anaplasmataceae and genus *Wolbachia* selected from the guts of *Atractomorpha sinensis* (As) were similar. The biomarkers screened in the guts of *Sphingonotus mongolicus* (Sm) were genus *Acinetobacter*, order Actinobacteria, phylum Actinobacteria, order Micrococcales and genus *Pantoea*, all of which had LDA values greater than 4. Figure 7 shows the relative abundance of each Biomarker. As can be clearly seen in panel A, the relative abundance of biomarkers in *Sphingonotus mongolicus* (Sm) was obviously high. In the three samples of the same grasshopper species, the relative abundance of Biomarker was different, which was the result of the differences between the samples.

### 3.2. Correlation Analysis of Bacteria

According to the abundance and change of each genus in each sample, Spearman rank correlation analysis was performed, and data of correlation > 0.1 and *p* < 0.05 were selected to construct a correlation network. Based on the analysis of the network diagram, the coexistence relationship of species in grasshopper intestinal samples could be obtained, and the interaction of species in the same environment and important model information could be obtained. Figure 8 shows the correlation analysis of the top 30 genera in abundances. *Klebsiella*, which has the highest relative abundance, had a significant positive correlation with *Enterobacter*, and had a significant negative correlation with *Wolbachia*, *Pantoea*, *Clostridium_sensu_stricto_1* and *Corynebacterium_1*.

### 3.3. Digestibility of Cellulose and Hemicellulose

After measurement and calculation, the cellulose content of wheat seedling was about 50.14%, hemicellulose content was about 8.39%. It was consistent with the cellulose content of Gramineae measured by Ye [24]. It was similar to the cellulose content of wheat straw, but significantly different to the hemicellulose content [25]. Table 4 shows the contents of cellulose and hemicellulose in the feces of four species of grasshoppers and the digestibility to cellulose and hemicellulose in wheat seedlings. The cellulose content of the feces of the *Sphingonotus mongolicus* (Sm) was 44.36% and *Trilophidia annulate* (Ta) was 41.54%. This indicated that *Trilophidia annulata* (Ta) had a slightly higher absorption of cellulose than that of the *Sphingonotus mongolicus* (Sm). Similarly, the absorption of cellulose by *Atractomorpha sinensis* (As) was higher than that by *Trilophidia annulata* (Ta), and by *Acrida cinerea* (Ac) was higher than that by *Atractomorpha sinensis* (As). In the same way, the content of hemicellulose in the feces of *Sphingonotus mongolicus* (Sm) reached 11.24%, ranking first among the four species of grasshoppers, followed by *Atractomorpha sinensis* (As) and *Trilophidia annulata* (Ta). Their fecal hemicellulose content was close. The hemicellulose content of the feces of *Acrida cinerea* (Ac) was 7.86%, ranking the last.

The cellulose digestibility of the four species of grasshoppers was higher than that of hemicellulose. Additionally, the digestibility of cellulose and hemicellulose of *Sphingonotus mongolicus* (Sm) were the highest, which were 67.91% and 47.51%, respectively. The results showed that the *Sphingonotus mongolicus* (Sm) had relatively high digestibility. The digestibility of cellulose from high to low were *Sphingonotus mongolicus* (67.91%), *Acrida cinerea* (56.97%), *Atractomorpha sinensis* (54.86%) and *Trilophidia annulata* (49.87%). The hemicellulose digestibility from high to low were *Sphingonotus mongolicus* (47.51%), *Acrida cinerea* (39.28%), *Trilophidia annulata* (19.25%) and *Atractomorpha sinensis* (17.77%). The digestibility of cellulose and hemicellulose had significant differences (*p* < 0.01), which may be due to the differences in the species and numbers of microorganisms.

### 3.4. Correlation Analysis of Intestinal Microorganism of Grasshopper with Digestibility of Cellulose and Hemicellulose

Cellulose and hemicellulose digestibility of four species of grasshoppers were determined, and Spearman correlations between them and gut microbes were analyzed. The results were shown in Figure 9, where CD represented cellulose digestibility and HD showed hemicellulose digestibility. Spearman correlation analysis showed that *Brevibacterium* (*p* < 0.01) and *Stenotrophomonas* (*p* < 0.05) were significantly correlated with cellulose digestibility. *Brevibacterium*, *Clavibacter*, *Microbacterium* and *Stenotrophomonas* were significantly correlated with the hemicellulose digestibility (*p* < 0.05). *Brevibacterium* can produce amylase [26]. Moreover, starch and cellulose were macromolecular polysaccharides composed of glucose. *Stenotrophomonas* could decompose xylan [27]. *Clavibacter* was a plant pathogen that destroyed plant cell walls by producing cellulases and pectinases [28]. This strain with cellulase activity isolated from insect guts included *Microbacterium* [29]. These also indirectly proved the reliability of the correlation analysis. The presence of these microorganisms helped grasshoppers digest plant cellulose and hemicellulose better.

## 4. Discussion

In this research, we used 16S rRNA gene high-throughput sequencing technology to analyze the bacterial diversity in the guts of four grasshopper species and determined the digestibility of cellulose and hemicellulose in those grasshoppers. We combined the analysis of the intestinal microbial diversity of *Acrida cinerea* (Ac), *Trilophidia annulata* (Ta), *Atractomorpha sinensis* (As) and *Sphingonotus mongolicus* (Sm) with their cellulose digestibility for the first time. This research showed that the composition of intestinal microorganisms of grasshoppers was diverse, which varied with different species, but there were still a small number of floras in common. There was a conserved core flora in different grasshopper species, which also indicated that the core flora had a symbiotic relationship with the grasshopper intestine and may play an important metabolic role in food digestion (cellulose degradation) and absorption. It laid a foundation for further research on the structure of the intestinal microorganism of grasshoppers, the relationship between microorganisms, the screening of microbial functional genes and the role of microorganisms in the life of grasshoppers.

Different living environments will lead to differences in the abundance and diversity of insect gut microbes [30,31]. Similarly, the gut microbial population of grasshoppers is also affected by relevant environmental conditions [5]. However, it is not clear how the living environment of grasshoppers affect their gut microbes. Yuan et al. confirmed that the gut bacterial structure of *G. molestacan* be influenced by the host plant [32]. Moro et al. showed that the diversity of gut microbes of the same species in regions was different [33]. Jesús M. et al. confirmed that different time scales strongly influence the diversity, composition and metabolic capabilities of *Brithys crini* gut microbial communities [34]. Huang et al. confirmed that both phylogeny and diet can impact the structure and composition of gut microbiomes [35]. The grasshoppers collected in this experiment were all adult, and the location and time were close to each other. To a large extent, the influence of time, environmental, climate and geographical conditions on the experimental results was avoided.

At the level of phylum, Proteobacteria accounted for the highest proportion, followed by Firmicutes. Muratore M. et al. found that there are bacterial phyla common to six grasshopper species from a coastal tallgrass prairie: Actinobacteria, Proteobacteria, Firmicutes, and to a lesser degree, Tenericutes [36]. Further, Wang et al. studied the gut microbial diversity of three species of grasshoppers, including *Oedaleus decorus asiaticus*, *Aiolopus tamulus* and *Shirakiacris shirakii*. Among them, Proteobacteria and Firmicutes were the most common. The intestinal microbial communities of the three species of grasshoppers are similar at the phylum level [22]. Mead et al. found that there were mainly four types of intestinal microbes, which were *Enterococcus* of Firmicutes, *Monserella*, *Pseudomonas* and *Enterobacteria* of Proteobacteria in the guts of *Melanoplus sanguinipes* [37]. In addition, using 16S rRNA gene sequencing, Schloss et al. found that the dominant intestinal phyla of *Saperda Vestita* was Proteobacteria [38]. Moreover, the largest relative proportion of the guts of the Mediterranean fruit fly was Enterobacteriaceae of Proteobacteria [39]. *Cnaphalocrocis medinalis* was the main pest of rice and the main dominant microflora in its larvae guts were Proteobacteria and Firmicutes [40]. Similarly, Kikuchi et al. found that the dominant microflora in the gut of *Riptortus cllavatus* were *Burkholderia* of Proteobacteria [41]. The above research results were consistent with this study. The abundance and structure of the intestinal microbes of these insects were different, but the dominant phyla were similar. At the genera level, *Klebsiella* accounted for the highest proportion in the intestinal microbes of the four species of grasshoppers, but the dominant genera were not the same. Barbosa et al. identified cellulase-producing bacteria by analyzing the 16S rDNA gene [42]. These strains were identified as *Klebsiella pneumoniae*, *Klebsiella* sp., and *Bacillus* sp. *Klebsiella pneumoniae* was the main cellulase-producing microorganism. In addition, Wang et al. found that *Klebsiella* accounted for the highest proportion of the microbial community in the three grasshopper species [22]. The specific role of *Klebsiella* in the guts of grasshoppers need to be further studied, but it was the common dominant bacteria in the guts of insects, and its important position cannot be ignored.

In this research, we found biomarkers with statistical differences between *Atractomorpha sinensis* (As) and *Sphingonotus mongolicus* (Sm) (Figure 7). The high abundance of *Acinetobacter*, *Pantoea*, and *Wolbachia* can be used as differential microorganisms to distinguish *Atractomorpha sinensis* (As) and *Sphingonotus mongolicus* (Sm). Hancock et al. found that *Wolbachia* could affect the reproduction of mosquitoes and reduce the spread of disease [43]. Further, a study on the brown planthopper showed that *Acinetobacter*, *Wolbachia* and *Staphylococcus* were significantly positively correlated with detoxification genes, that is, these symbiotic bacteria were involved in the metabolism of insecticides in the guts of the brown planthopper, which had positive significance for pest control [44]. *Acinetobacter* had esterase activity [45], which may also be related to nutrient metabolism of host insects [40]. *Pantoea* could provide vitamins and amino acids for host insects [46]. Therefore, gut microbes are closely related to the life activities of the host, differential microorganisms can be used in subsequent studies to explore their functions.

The main place where most bacteria in insect guts exist is the mid-hindgut [47]. In this research, we selected the mid-hindgut of grasshoppers as the experimental material, and the results proved that *Klebsiella* sp. were the common dominant bacteria in four species of grasshoppers. *Klebsiella* belongs to the Enterobacteriaceae. Smith et al. found that Enterobacteriaceae mainly reside in the hindgut and are involved in carbohydrate metabolism [48]. It has been reported that in the gut of *Bactrocera Oleae*, *Klebsiella* and *Enterobacter* were harmful for the host [49]. *Klebsiell oxytoca* in the gut of fruit fly delayed the emergence of parasitic wasps [50]. This suggested that *Klebsiella* and *Enterobacter* had some positive correlation and worked together in the hosts. However, Gao et al. found that *Klebsiella* can promote the growth and development of *Drosophila suzukii* to a certain extent [51]. Moreover, the *Klebsiella* isolated from the oral secretion of fall armyworm could down-regulate the activity of peroxidase and up-regulate the activity of trypsin inhibitor in tomato, thereby reducing the ability of tomato to resist pests [52]. The *Klebsiella* isolated from the larvae of *Dendrolimus kikuchii* could produce lipase [53].

In the determination of cellulose digestibility, adding sulfuric acid produced a large amount of heat, reducing the accuracy of the experimental results. The ice bath could effectively solve this problem and ensure the accuracy of the results. The results showed that the digestibility of cellulose was higher than that of hemicellulose. The digestibility of cellulose and hemicellulose varied greatly, which was related to the structure of cellulose and hemicellulose. The chemical structure of cellulose had high degree of polymerization, and the hydrogen bonding force between molecules determined that it was difficult to degrade [24]. Cellulose and hemicellulose had different decomposition products and different proportions in plants [54]. Consequently, the digestibility of grasshoppers was significantly different. The decomposition of cellulose and hemicellulose required the cooperation of a variety of microorganisms. The microorganisms that secreting cellulase may not participate in the breakdown of hemicellulose. Therefore, the number of microorganisms secreting cellulase and hemicellulose would affect the digestibility. The differences in the cellulose digestibility may be due to the differences in the amount of cellulase in the guts of different grasshoppers. The grasshoppers with high cellulose digestibility had a large number of microorganisms in their guts that can decompose cellulose and secrete a large amount of cellulase with high activity. The same was true for hemicellulose. Tian and Ba found that the cellulose digestibility of the rumen fluid of Tibetan sheep to the highland barley straw is 25.8% [55]. Li et al. studied the digestibility of sheep to corn stalks treated in different ways, and the results showed that the digestibility of crude fiber was 34.21–61.21% [56]. Further, Zhang et al. found that the digestibility of neutral detergent fiber and acid detergent fiber to wheat straw were 28.5–30.9% and 29.1–37.0%, respectively [57]. The cellulose digestibility in this research was close to that of mammals, and far higher than that of *Locusta migratoria manilensis*. This result may be due to different feeding materials. However, the digestion and utilization of cellulose and hemicellulose in the four species of grasshoppers were at high level, which might be related to the microorganisms in the gut. The breakdown of cellulose and hemicellulose requires the participation of enzymes. Additionally, the secretion of these enzymes requires the cooperation of a large number of microorganisms. However, in the guts of grasshoppers, which microorganism had the ability to decompose cellulose and hemicellulose and what was their specific roles in the decomposition process still need further research. *Bacillus licheniformis*, *O. intermedium*, and *M. paludicola* were isolated from the gut contents of termites (*Microcerotermes diversus*) as described previously [58]. They have high cellulose degradability. Kundu found 15 hemicellulolytic microbes in the guts of termites [59]. Similarly, Huang et al. isolated *Cellulomonas* sp. h9 from the intestinal tract of larvae of *Protaetia brevitarsis* [60]. It provides a research basis for the isolation of cellulose-degrading bacteria in the intestines of grasshoppers, which can be further studied. Some microorganisms related to cellulose and hemicellulose was obtained from Spearman correlation analysis. However, what role they play in the catabolism of cellulose and hemicellulose remains to be further verified. The digestibility of cellulose and hemicellulose of the four species of grasshoppers are high, and they have potential value as bioreactors for lignocellulose decomposition.

## 5. Conclusions

In conclusion, 16S rRNA gene sequences was used to determine the bacterial diversity of *Acrida cinerea*, *Trilophidia annulata*, *Atractomorpha sinensis* and *Sphingonotus mongolicus*, and correlation analysis was performed between the intestinal microbes of grasshoppers and the digestibility of cellulose and hemicellulose.

The diversity and abundance of intestinal microorganisms were different among all species, but there was no significant difference. *Acrida cinerea* had the highest bacterial species richness, and *Sphingonotus mongolicus* had the highest bacterial diversity. Proteobacteria and Firmicutes are the dominant bacteria in the intestinal microbial communities of the four grasshopper species. The dominant genera of different species of grasshoppers are different, and the common dominant species is *Klebsiella*. The intestinal microflora structure varied among the different species of grasshoppers, with the intestinal microflora structure of *Acrida cinerea* and *Atractomorpha sinensis* being the most similar. In addition, *Sphingonotus mongolicus* had the highest digestibility. The digestibility of cellulose was significantly different among species, as was the digestibility of hemicellulose. The digestibility of cellulose was higher than that of hemicellulose. Further, Spearman correlation analysis showed that *Brevibacterium* and *Stenotrophomonas* were significantly correlated with the cellulose digestibility. *Brevibacterium*, *Clavibacter*, *Microbacterium* and *Stenotrophomonas* were significantly correlated with the hemicellulose digestibility. The microorganisms mentioned above can be used as back-up to break down cellulose and hemicellulose.

Increasing the understanding of the structure and function of the grasshopper intestinal microflora will facilitate further research and the utilization of intestinal microorganisms in the future and contribute to the development of grasshoppers as a cellulose degradation bioreactors. Meanwhile, it provides a new idea for the decomposition and utilization of straw in agriculture and animal husbandry, which has important theoretical and practical significance.

## Figures and Tables

**Figure 1 insects-13-00432-f001:**
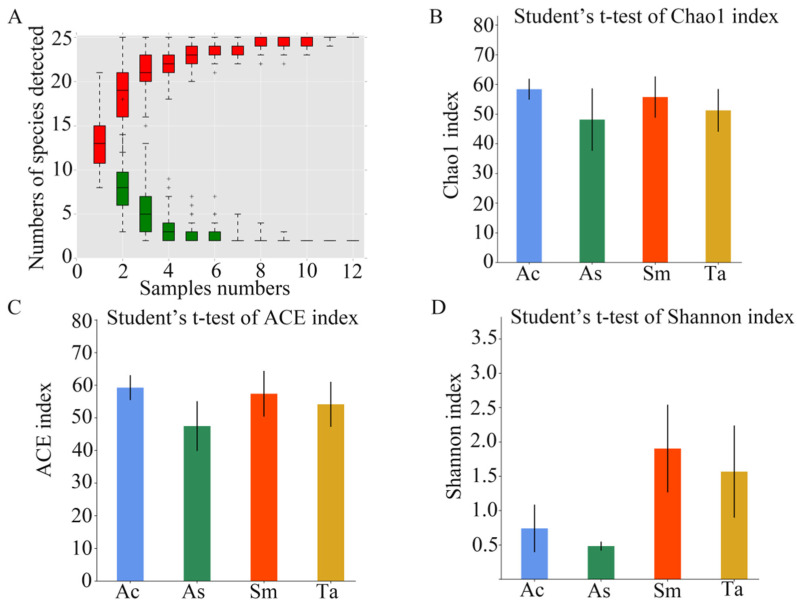
The results of α-diversity analysis. (**A**) Species discovery curve. (A single red box in this figure represents the total number of species detected in randomly selected samples. The cumulative curve is composed of the totality of red boxes, which represents the rate of new species appearing under continuous sampling; a single green box in this figure represents the number of common species detected in a given number of samples. The set of green boxes form the common quantity curve, which represents the rate of common species detected under continuous sampling). (**B**) Chao1 index of the four species of grasshoppers. (**C**) ACE index of the four species of grasshoppers. (**D**) Shannon index of the four species of grasshoppers. Ac, *Acridia chinensis*; As, *Atractomorpha sinensis*; Sm, *Sphingonotus mongolicus*; Ta, *Trilophidia annulate*.

**Figure 2 insects-13-00432-f002:**
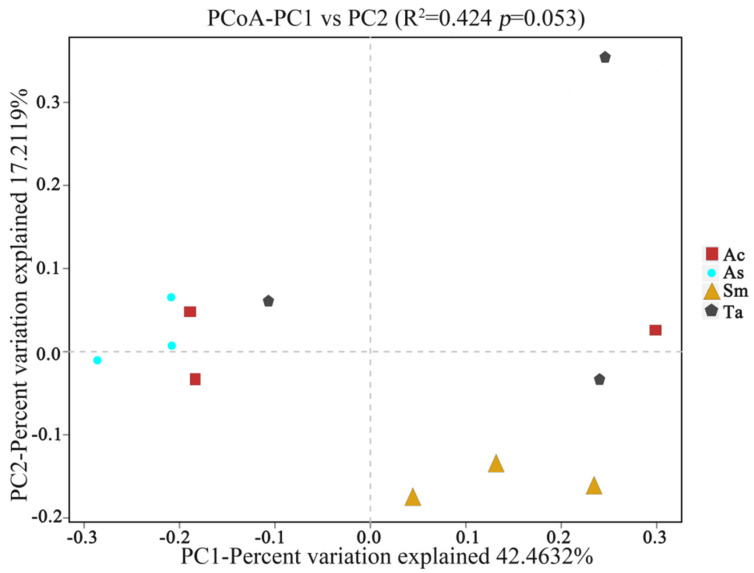
Principal coordinate analysis (PCoA) between four species of grasshoppers. Ac: red rectangle; As: blue circle; Sm: yellow triangle; Ta: gray polygon. The horizontal and vertical coordinates are the two eigenvalues that cause the largest difference between samples, and the main influence degree is reflected in the form of percentage. Ac, *Acridia chinensis*; As, *Atractomorpha sinensis*; Sm, *Sphingonotus mongolicus*; Ta, *Trilophidia annulate*.

**Figure 3 insects-13-00432-f003:**
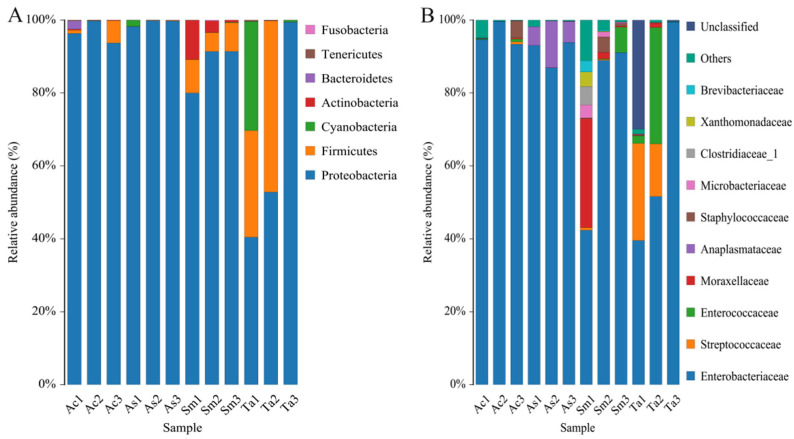
Bacterial abundance histogram at (**A**) the phylum level and (**B**) the family level. Each color represents a species, and the height of the color block indicates the proportion of the species in relative abundance. The top 10 genera in relative abundance were shown in Figure 3B. “Others” represented the remaining. “Unclassified” represented OTUs that were not commented. Ac, *Acridia chinensis*; As, *Atractomorpha sinensis*; Sm, *Sphingonotus mongolicus*; Ta, *Trilophidia annulate*.

**Figure 4 insects-13-00432-f004:**
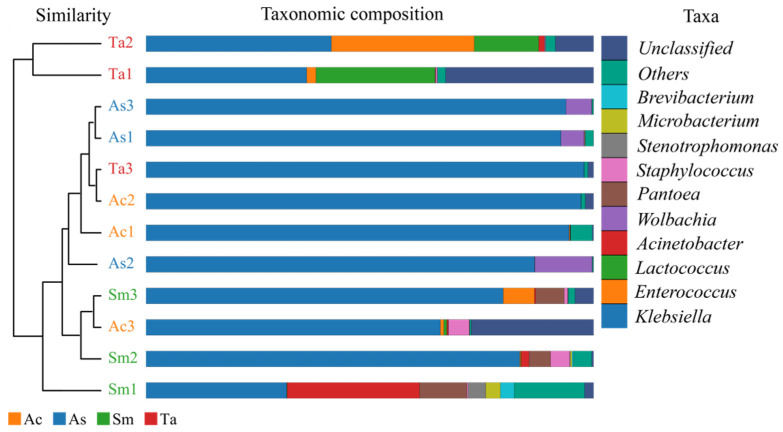
UPGMA cluster tree combined with histogram. The clustering tree and the abundance histogram are shown on the left and on the right, respectively. The top 10 genera in relative abundance were shown here. “Others” represented the remaining. “Unclassified” represented OTUs that were not commented. Ac, *Acridia chinensis*; As, *Atractomorpha sinensis*; Sm, *Sphingonotus mongolicus*; Ta, *Trilophidia annulate*.

**Figure 5 insects-13-00432-f005:**
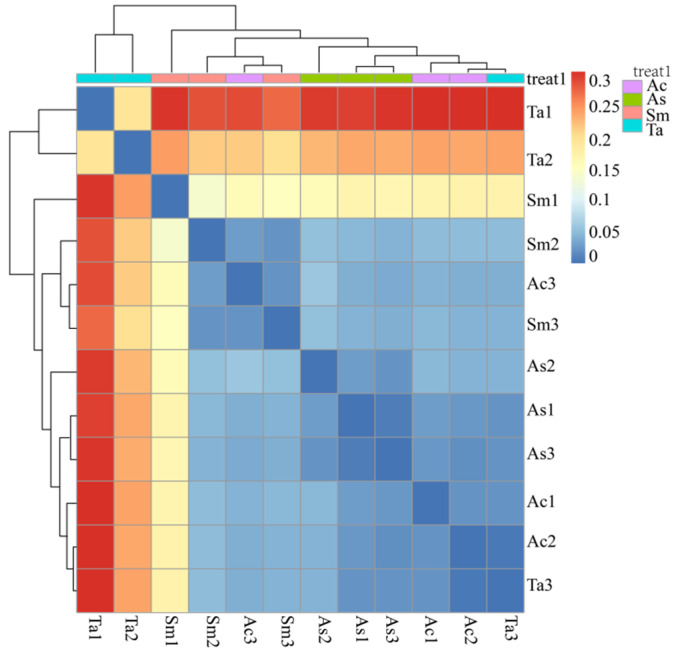
Heatmap of each sample at the OTU classification level. Dendrograms of hierarchical cluster analysis samples are shown on the left and at the top, respectively. The color gradient from blue to red indicated that the distance between the samples was from close to far. Ac, *Acridia chinensis*; As, *Atractomorpha sinensis*; Sm, *Sphingonotus mongolicus*; Ta, *Trilophidia annulate*.

**Figure 6 insects-13-00432-f006:**
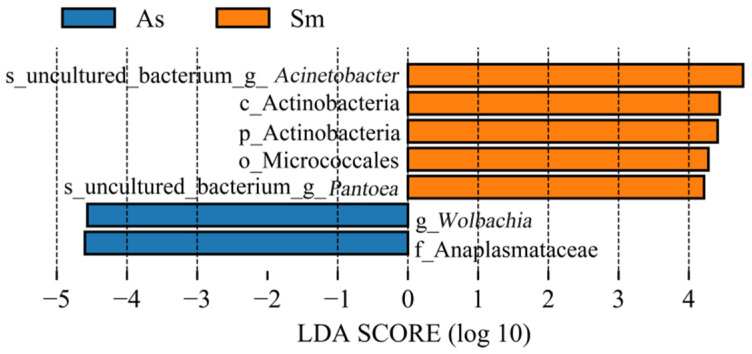
Bacterial taxa with linear discriminant analysis (LDA) score greater than four in the gut microbiota of different grasshopper. As, *Atractomorpha sinensis*; Sm, *Sphingonotus mongolicus*.

**Figure 7 insects-13-00432-f007:**
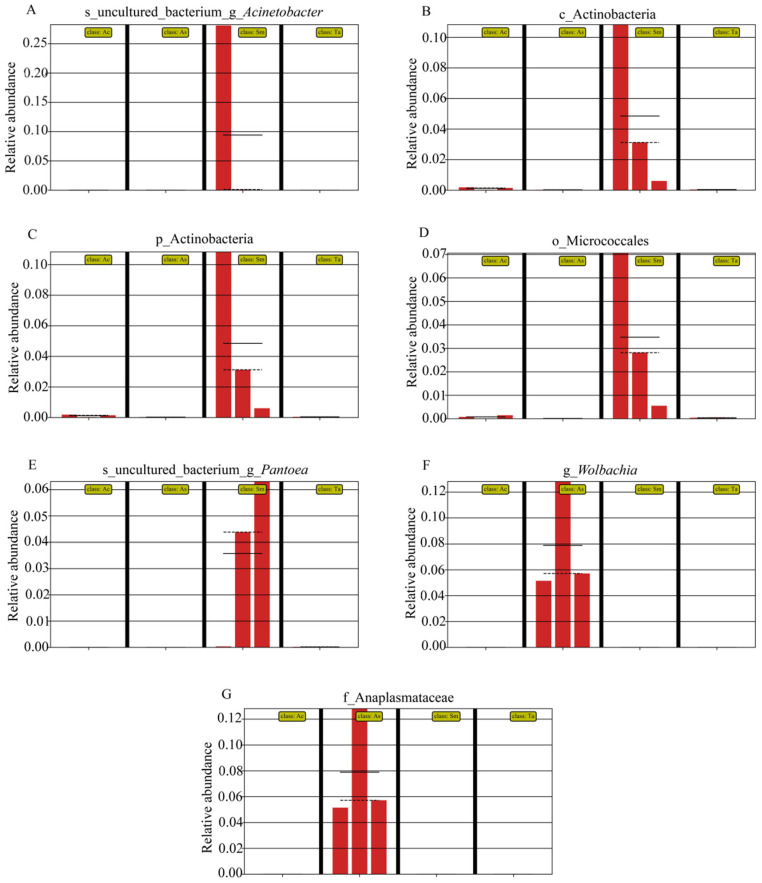
Abundance histogram of bacterial taxa with linear discriminant analysis (LDA) score greater than four in the gut microbiota. Different groups are separated by black bold solid lines. The solid line in the histogram of each group represents the average value of the expression amount of the reorganized sample, and the dotted line represents the median value of the expression amount of the group of samples. (**A**) s_uncultured_bacterium_g_*Acinetobacter*. (**B**) c_Actinobacteria. (**C**) p_ Actinobacteria. (**D**) o_Micrococcales. (**E**) s_uncultured_bacterium_g_*Pantoca*. (**F**) g_*Wolbachia*. (**G**) f_Anaplasmataceae.

**Figure 8 insects-13-00432-f008:**
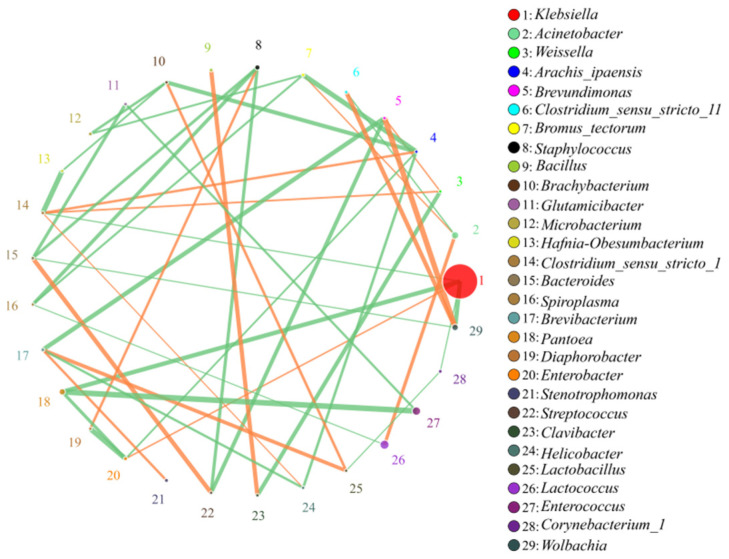
Correlation analysis of microorganisms at the genus level. The circles represented the genera, the size of the circles represented the average abundance, the lines represented the correlation between two species, the thickness and thinness of the lines represented the strength of the correlation, orange represented a positive correlation and green represented negative correlation. The correlation analysis of the top 30 genera in abundances are shown on the right.

**Figure 9 insects-13-00432-f009:**
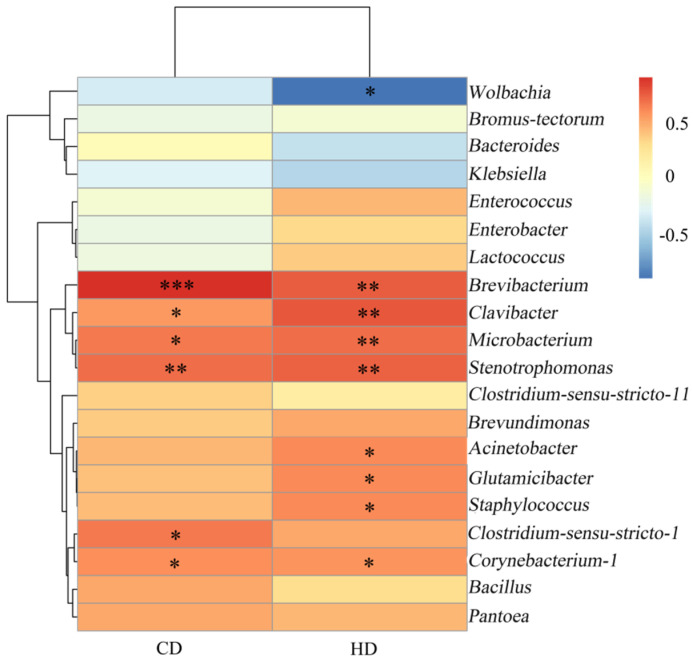
Heatmap of the correlation between digestibility and bacterial abundance. Dendrograms of hierarchical cluster analysis grouping genera is shown on the left. * There is a significant correlation of 5% between digestibility and bacteria. ** There is a significant correlation of 1% between digestibility and bacteria. *** There is a significant correlation of 0.1% between digestibility and bacteria. CD, cellulose digestibility; HD, hemicellulose digestibility.

**Table 1 insects-13-00432-t001:** Basic information of experimental specimens.

Species	Sample Code	Collection Date	Locality
*Acrida cinerea*	Ac1	July 2018	Baoding, China
	Ac2	July 2018	
	Ac3	October 2018	
*Trilophidia annulate*	Ta1	October 2018	Baoding, China
	Ta2	October 2018	
	Ta3	July 2018	
*Sphingonotus mongolicus*	Sm1	October 2018	Baoding, China
	Sm2	July 2018	
	Sm3	July 2018	
*Atractomorpha sinensis*	As1	July 2018	Baoding, China
	As2	July 2018	
	As3	July 2018	

**Table 2 insects-13-00432-t002:** Species statistics table of each grade of the sample.

Sample	Kingdom	Phylum	Class	Order	Family	Genus
Ac1	1	6	10	17	34	36
Ac2	1	6	10	15	28	32
Ac3	1	6	9	13	24	28
As1	1	5	9	14	23	26
As2	1	5	9	15	28	29
As3	1	5	10	14	24	28
Sm1	1	6	10	18	33	41
Sm2	1	6	10	18	32	37
Sm3	1	5	8	15	24	28
Ta1	1	6	10	16	30	35
Ta2	1	5	6	9	18	22
Ta3	1	6	9	16	26	29
Total	1	7	12	21	42	55

**Table 3 insects-13-00432-t003:** The average value of the Alpha diversity index of each sample.

Species	Sample ID	ACE	Chao1	Shannon	Coverage
*Acrida cinerea*	Ac	59.2633	58.3929	0.7407	0.9997
*Atractomorpha sinensis*	As	47.4831	48.1513	0.4829	0.9998
*Sphingonotus mongolicus*	Sm	57.3990	55.7500	1.9040	0.9999
*Trilophidia annulate*	Ta	54.1572	51.2593	1.5683	0.9998

**Table 4 insects-13-00432-t004:** The content and digestibility of cellulose and hemicellulose.

Species of Grasshopper	Sample ID	Cellulose Content in Feces (%)	Cellulose Digestibility (%)	Hemicellulose Content in Feces (%)	Hemicellulose Digestibility (%)
*Acrida cinerea*	Ac	33.28 ± 0.02	56.97 ± 0.09	7.86 ± 0.01	39.28 ± 0.12
*Atractomorpha sinensis*	As	37.29 ± 0.02	54.86 ± 0.06	11.37 ± 0.01	17.77 ± 0.10
*Sphingonotus mongolicus*	Sm	44.36 ± 0.03	67.91 ± 0.08	12.14 ± 0.01	47.51 ± 0.12
*Trilophidia annulata*	Ta	41.54 ± 0.04	49.87 ± 0.06	11.20 ± 0.02	19.25 ± 0.09

## Data Availability

The data presented in this study are available in article here.

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
