# Peer review of "Analysis of Intestinal Microbial Diversity of Four Species of Grasshoppers and Determination of Cellulose Digestibility"

_insects, 2022, doi:10.3390/insects13050432_

Round 1

Reviewer 1 Report

This study investigated the gut bacterial community structure of 4 grasshopper species: Acrida cinerea, Trilophidia annulata, Atractomorpha sinensis, and Sphingonotus mongolicus. This study is of interest to the growing community of insect-microbe associations researchers and the need for description of more Acridid species. Unfortunately, this paper needs substantial improvement to be publishable. I think the study is meritorious but has been underappreciated and under analyzed. 

First of all the introduction and discussion lack critical literature review. Important works from Lavy et al. 2019, 2022, Smith et al. 2017 explore gut microbiota from other Orthopterans that would be important for comparison and analysis. 

Secondly, the methods are not written in such a way for someone to repeat them nor with the appropriate vocabulary. Grasshoppers do not have stomachs or chests. I have no idea how the grasshoppers were held for the digestion experiment (3 days + 2 days is not one week).  

In the results, several statements are made without clear comparisons. For instance at the top of page 6, " Two-way ANOVA showed that there was no significant difference..." but in the next paragraph the authors mention that bacterial genera >1% were "also different" -- than what? Similarly, the next paragraph at the bottom of page 6 begins that the LefSe analysis was "to find the different microorganisms" what does this mean? Significantly different? I thought the ANOVA demonstrated there were no differences?

The figure captions throughout the paper must be improved. They should include all the relevant information to evaluate the figure independent of the text. Figure captions MUST include all abbreviations. How does figure 2 add to the paper compared to Figure 1? Where are the data for TA and AC? Why are they not included in the LefSe analysis when they were also sequenced?

The analysis of the alpha and beta diversity data is superficial. What does these numbers mean? Why are you reporting them? Why are they important to this discussion? What does it mean that the p values were low if the PCoA plot doesn't seem striking to the authors. Additionally, figure 4 is impossible to interpret in black/white which should be considered by the authors. Does figure 5 need to be included? Could it replace Figure 1? Which grasshopper(s) are being evaluated in figure 8??

With all of this in mind, I cannot recommend this paper for publication in its current form 

Reviewer 2 Report

Comments and Suggestions for Authors:

The manuscript entitled " Analysis of intestinal microbial diversity of four species of grasshoppers and determination of cellulose digestibility" addressed the gut microbial community structure of the grasshopper and observed the relationship between gut microbial community and cellulose digestibility. The methods are well adequate, and the results are well described. However, there are a few corrections and comments that need to be addressed by the authors. Please update all the figures and improve the writing as mentioned below. Thank you.

Abstract: Line 4: “In this Paper”, Should be “In this research”

Figure 1-6 & 8: Legend should be clearer and more readable (could not see properly)

Figure 7: the figure is not clearly visible and could not be able to read the legend. Please change the legend and update the figure which will be easy for the reader to see.

Introduction:

1st Paragraph: Please include this correction:

“In the gut of insects, there are many types of microorganisms available, which can be divided into resident microflora and passing microflora.”

Rewrite the following sentences:

“Therefore, the study of insect gut microorganisms is of great significance”.

“There are few researches on the degradation mechanism of cellulose plants by grasshoppers, the digestibility of cellulose and hemicellulose, and the relationship between this digestibility and intestinal microorganisms.”

Discussion:

1st paragraph and other paragraphs of the discussion section, need to add few Transitional Words and Phrases for example: further, moreover, in addition etc.

Example:

“However, it is not clear how the living environment of grasshoppers affects their gut microbes. Yuan et al. confirmed that the gut bacterial structure of G. molestacan be influenced by the host plant [25]. Furthermore, Moro et al. showed that the diversity of gut microbes of the same species in different regions is different [26]. In addition, Huang et al. confirmed that both phylogeny and diet can impact the structure and composition of gut microbiomes [27].”

2nd paragraph:

Include gap between two lines:

“Acrida cinerea, Trilophidia annulata, Atractomorpha sinensis and Sphingonotus mongolicus in the same geographical environment were used in the study.12 samples from 4 species of grasshopper were annotated to 7 phyla, 12 classes, 21 orders, 42 families and 54 genera.”

4th Paragraph:

Please rewrite this sentence to make it more understandable and easier to read

“In this study, microorganisms that can be used as biomarkers can be obtained from Figure 3, which can not only be used to distinguish two species of grasshoppers or samples, but also can be used in subsequent studies to explore their functions.”

Conclusion:

The conclusion has been poorly written (too big and vague) and needs to rewrite the paragraph
